# *S. mutans* Antisense *vicK* RNA Over-Expression Plus Antibacterial Dimethylaminohexadecyl Methacrylate Suppresses Oral Biofilms and Protects Enamel Hardness in Extracted Human Teeth

**DOI:** 10.3390/pathogens13080707

**Published:** 2024-08-21

**Authors:** Shuang Yu, Mengmeng Xu, Zheng Wang, Yang Deng, Hockin H. K. Xu, Michael D. Weir, Negar Homayounfar, Guadalupe Garcia Fay, Hong Chen, Deqin Yang

**Affiliations:** 1Department of Endodontics, Stomatological Hospital of Chongqing Medical University, Chongqing 404100, China; 2Chongqing Key Laboratory of Oral Diseases and Biomedical Sciences, Stomatological Hospital of Chongqing Medical University, Chongqing 404100, China; 3Chongqing Municipal Key Laboratory of Oral Biomedical Engineering of Higher Education, Chongqing 404100, China; 4Chongqing Key Laboratory of Oral Diseases and Biomedical Sciences, 426 Songshi North Road, Yubei Distrinct, Chongqing 401147, China; 5Department of Biomaterials and Regenerative Dental Medicine, University of Maryland Dental School, Baltimore, MD 21201, USA; 6Department of Advanced Oral Sciences and Therapeutics, University of Maryland Dental School, Baltimore, MD 21201, USA

**Keywords:** antisense *vicK* RNA (AS*vicK*), dimethylaminohexadecyl methacrylate (DMAHDM), dental caries, biofilm formation, enamel demineralization

## Abstract

*Streptococcus mutans* (*S. mutans*) antisense *vicK* RNA (AS*vicK*) is a non-coding RNA that regulates cariogenic virulence and metabolic activity. Dimethylaminohexadecyl methacrylate (DMAHDM), a quaternary ammonium methacrylate used in dental materials, has strong antibacterial activity. This study examined the effects of *S. mutans* AS*vicK* on DMAHDM susceptibility and their combined impact on inhibiting *S. mutans* biofilm formation and protecting enamel hardness. The parent *S. mutans* UA159 and AS*vicK* overexpressing *S. mutans* (AS*vicK*) were tested. The minimum inhibitory concentration (MIC) and minimum bactericidal concentrations for planktonic bacteria (MBC-P) and biofilms (MBC-B) were measured. As the AS*vicK* MBC-B was 175 μg/mL, live/dead staining, metabolic activity (MTT), colony-forming units (CFUs), biofilm biomass, polysaccharide, and lactic acid production were investigated at 175 μg/mL and 87.5 μg/mL. The MIC, MBC-P, and MBC-B values for DMAHDM for the AS*vicK* strain were half those of the UA159 strain. In addition, combining *S. mutans* AS*vicK* with DMAHDM resulted in a significant 4-log CFU reduction (*p* < 0.05), with notable decreases in polysaccharide levels and lactic acid production. In the in vitro cariogenic model, the combination achieved the highest enamel hardness at 67.1% of sound enamel, while UA159 without DMAHDM had the lowest at 16.4% (*p* < 0.05). Thus, *S. mutans* AS*vicK* enhanced DMAHDM susceptibility, and their combination effectively inhibited biofilm formation and minimized enamel demineralization. The *S. mutans* AS*vicK* + DMAHDM combination shows great potential for anti-caries dental applications.

## 1. Introduction

Dental caries is a multifactorial, biofilm-mediated, chronic oral disease [1,2,3]. *Streptococcus mutans* (*S. mutans*) is considered the most significant etiological pathogen due to its exceptional ability to form biofilms [4]. Within the plaque biofilm, *S. mutans* utilizes polysaccharides and produces organic acids, primarily lactic acid, which accounts for 70% of the acids in the oral biofilm. These acids lead to tooth demineralization and, ultimately, dental caries [5,6,7]. Thereby, inhibiting biofilm formation presents a promising strategy to mitigate the cariogenic potential of *S. mutans* and prevent dental caries.

Multiple genes and pathways are involved in biofilm formation and acid production. A previous study aimed to reduce cariogenic virulence at the genetic level by targeting two-component signal transduction systems (TCSs) [8]. In *S. mutans*, the VicRK system is an essential TCS that influences biofilm formation, acid production, and exopolysaccharide (EPS) synthesis by regulating several virulence-associated genes [9]. The VicRK system typically comprises two regulatory elements: a membrane-located histidine kinase (VicK) and a cytoplasmic response regulator (VicR) [10]. Upon the detection of a specific environmental stimulus, a sensor histidine kinase-VicK initiates a histidine phosphorelay to VicR. VicR then positively regulates downstream gene expression, including glucosyltransferase B (*gtfB*), glucosyltransferase C (*gtfC*), glucosyltransferase D (*gtfD*), and fructosyltransferase (ftf) [11]. Therefore, modulating VicK activity could weaken the cariogenicity of *S. mutans* and help prevent caries.

Biofilm growth and maturation is intricately linked to EPS synthesis, which provides nutrients and protective barrier function against drugs, enhancing resistance to antimicrobial agents [8]. Targeting EPS decomposition is an effective method to control plaque and reduce its cariogenicity. Antisense RNAs (asRNAs), a class of non-coding RNAs (ncRNAs), modulate processes ranging from chromatin changes to RNA editing [12]. In our previous studies, we identified an antisense RNA in *S. mutans* with reverse phosphorelay complementarity to *vicK* mRNA, known as antisense *vicK* RNA (AS*vicK*). AS*vicK* specifically binds to msRNA1657 and recruits RNase III, co-regulating the expression of *vicK* at both transcriptional and post-transcriptional levels [13,14]. This regulation influences biofilm and EPS formation, thereby attenuating the cariogenic potential of *S. mutans*.

Quaternary ammonium methacrylates (QAMs) can be co-polymerized and covalently bonded within dental resin, providing long-term broad-spectrum antibacterial effects through contact inhibition [15,16,17]. The antibacterial effectiveness of QAMs is influenced by the length of the N-alkyl chain [18]. Previous research has demonstrated that increasing the chain length from 3 to 16 significantly enhances antibacterial efficacy, although effectiveness decreases at a chain length of 18 [19]. Various quaternary ammonium monomers, including 12-methylacryloxydodecylpyridinebromide (MDPB), quaternary ammonium polyethylenimine (QPEI), dimethylaminododecylmethacrylate (DMADDM), and dimethylaminohexadecylmethacrylate (DMAHDM), have been synthesized and incorporated into dental materials for research purposes [20,21,22]. Among these, DMAHDM is notable for its long alkyl chain, excellent biocompatibility, minimal impact on material mechanical properties, and low cytotoxicity. These characteristics make it a promising antibacterial monomer for use in resin-based dental materials, particularly when combined with components such as 2-methacryloyloxyethyl phosphorylcholine (MPC), nanoparticles of calcium fluoride (nCaF_2_), and nanoparticles of amorphous calcium phosphate (NACP) [23,24,25,26,27,28].

In the oral environment, a dynamic equilibrium exists between demineralization and remineralization at the tooth-biofilm-saliva interface [29]. Numerous in vitro biofilm models have been developed to investigate the efficacy of antibacterial materials in preventing enamel demineralization [30,31,32]. These models operate under highly standardized conditions within a set timeframe. In this study, we used a strain carrying AS*vicK* and established an in vitro cariogenic model to explore the combined effects of *S. mutans* AS*vicK* and DMAHDM on inhibiting enamel demineralization. Therefore, the objectives of this study were to investigate for the first time: (1) the effects of *S. mutans* AS*vicK* on DMAHDM susceptibility; (2) the combined impact of *S. mutans* AS*vicK* and DMAHDM on inhibiting biofilm formation and enamel demineralization.

## 2. Materials and Methods

### 2.1. Synthesis of DMAHDM

DMAHDM was synthesized following previously established methods [3,33]. Briefly, 10 mmol of 2-(dimethylamino) ethyl methacrylate (DMAEMA, Sigma-Aldrich, St. Louis, MO, USA), 10 mmol of 1-bromododecane (BDD, TCI America, Portland, OR, USA), and 3 g of ethanol were mixed and stirred at 70 °C for 24 h. The ethanol solvent was then evaporated, leaving behind DMAHDM [3]. The sample was subsequently subjected to vacuum to remove any remaining solvents, unreacted components, and impurities. The chemical structure of the reaction product was confirmed using Fourier-transform infrared spectroscopy (FTIR) and hydrogen-1 nuclear magnetic resonance (^1^H-NMR). Both FTIR and ^1^H-NMR analyses detected no impurities, indicating a purity close to 100%.

### 2.2. Strain Verification and Biofilm Formation

This study received approval from the Ethics Committee of the School of Stomatology, Chongqing Medical University, for the study titled “Antisense *vicK* Long Non-coding RNA Regulation of Quaternary Ammonium Salt Resistance in *Streptococcus mutans* and Its Mechanism Study” (CQHS-REC-2020-064). All bacterial strains and primer sequences used are listed in Table 1 and Table 2. The parent *S. mutans* UA159 (ATCC 700610) was obtained from the State Key Laboratory of Oral Diseases (Sichuan University, Chengdu, China). The AS*vicK* overexpression strain was constructed from *S. mutans* UA159 by introducing the AS*vicK* overexpression plasmid pDL278 [13]. Verification of the UA159 and AS*vicK* strains was performed using quantitative real-time PCR (qRT-PCR) with spectinomycin at a concentration of 1 mg/mL [13,14].

The parental *S. mutans* were cultured in brain-heart infusion broth (BHI; Difco, Sparks, MD, USA) without antibiotics and incubated at 37 °C with 90% N_2_ and 10% CO_2_ [34]. For the AS*vicK* strain, 1 mg/mL spectinomycin was added to the BHI medium. Overnight bacterial suspensions were added to fresh BHIS (BHI with 1% sucrose) at a concentration of 2 × 10^6^ CFU/mL, determined by OD_600nm_ = 0.10 [34]. The suspensions were distributed into 24-well plates and incubated for 24 h. DMAHDM was dissolved in double-distilled water (ddH_2_O) to form a clear stock solution at 7 mg/mL. For the experiments, the solution was diluted to concentrations of 175 μg/mL and 87.5 μg/mL. After the initial incubation, the 24-well plates were washed twice with phosphate-buffered saline (PBS) to remove the biofilms from the well surface. Fresh BHI containing DMAHDM at specified concentrations was then added to the plates. The mass fractions refer to the ratio of the DMAHDM mass to the total solution mass. The cultures were incubated for an additional 24 h, resulting in 48-h biofilms for subsequent experiments.

**Table 2 pathogens-13-00707-t002:** Sequences of primers used for quantitative real-time PCR (qRT-PCR) analysis.

Primers	Sequence 5′-3′ (Forward/Reverse)	Source or Reference
*gyrA*	5′-ATTGTTGCTCGGGCTCTTCCAG-3′/ 5′-ATGCGGCTTGTCAGGAGTAACC-3′	[35]
AS*vicK*	5′-GTTGATTGCTGACTTTGAATTTGA-3′/ 5′-GCGTATGATTACTGATTTACTTAGC-3′	[13]
*vicK*	5′-CACTTTACGCATTCGTTTTGCC-3′/ 5′-CGTTCTTCTTTTTCCTGTTCGGTC-3′	[35]
*gtfB*	5′-ACACTTTCGGGTGGCTTG-3′/ 5′-GCTTAGATGTCACTTCGGTTG-3′	[35]
*gtfC*	5′-CCAAAATGGTATTATGGCTGTCG-3′/ 5′-TGAGTCTCTATCAAAGTAACGCAG-3′	[35]
*gtfD*	5′-AATGAAATTCGCAGCGGACTTGAG-3′/ 5′-TTAGCCTGACGCATGTCTTCATTGTA-3′	[35]
*ftf*	5′-ATTGGCGAACGGCGACTTACTC-3′/ 5′-CCTGCGACTTCATTACGATTGGTC-3′	[35]

### 2.3. Determining MIC, MBC-P, and MBC-B of DMAHDM

The minimum inhibitory concentration (MIC) is the lowest concentration of an antimicrobial agent that inhibits visible microbial growth [36]. Measurements were performed using the serial, two-fold microdilution method described in a previous study [37]. Overnight bacterial suspensions were adjusted to a concentration of 2 × 10^6^ CFU/mL. DMAHDM at 112 μg/mL was added to the first well and then serially diluted two-fold in the medium to obtain a series of concentration gradients [37]. Each solution was prepared in a 96-well plate with 100 μL per well [37]. Following incubation at 37 °C with 90% N_2_ and 10% CO_2_ for 24 h, the MIC was determined by measuring the OD_600nm_. The lowest concentration at which the OD_600nm_ showed the most significant decrease was recorded as the MIC. The minimum bactericidal concentration of planktonic bacteria (MBC-P) is the lowest concentration of an antimicrobial agent that kills 99.9% of the initial bacterial inoculum, determined by plate counting viable bacteria [38]. After MIC determination, 10 μL aliquots of bacterial suspensions from wells with a significant decrease in OD_600nm_ were taken. These aliquots were then plated onto BHI agar plates. The plates were incubated anaerobically (90% N_2_ and 10% CO_2_) for 48 h at 37 °C. Following CFU counting, the MBC-P was determined [34].

The minimum bactericidal concentration of biofilms (MBC-B) was determined as described in a previous study [39]. Overnight bacterial suspensions were inoculated into a 96-well plate, and biofilms were allowed to form for 24 h at 37 °C with 90% N_2_ and 10% CO_2_ [39]. Subsequently, the BHIS medium was replaced with serial dilutions of fresh BHIS containing DMAHDM, starting at a concentration of 350 μg/mL [37]. The plates were then incubated for an additional 24 h under the same conditions. The biofilm was scraped from the bottom of the wells into tubes containing 1 mL of sterile PBS using pipette tips and repeated pipetting, followed by vortexing at 2400 rpm for 30 s using a vortex mixer (Fisher, Pittsburgh, PA, USA) [40,41]. Then, 10 μL of the bacterial suspensions were plated onto BHI agar to determine the drug concentration that killed 99.9% of the initial biofilm bacteria. After CFU counting, the MBC-B was determined.

### 2.4. Growth Curves

Following a previous study, the overnight bacterial suspensions were adjusted to a concentration of 2 × 10^6^ CFU/mL [42]. DMAHDM was dissolved in sterile ddH_2_O and then diluted with culture medium to concentrations of 0.875 μg/mL, 1.75 μg/mL, 3.5 μg/mL, 7 μg/mL, and 14 μg/mL for both UA159 and AS*vicK* strains. The bacteria were cultured at 37 °C with 90% N_2_ and 10% CO_2_. The OD_600nm_ values were recorded at 2-h intervals during the 22-h growth period using a microplate reader (SpectraMax M5, Molecular Devices, Sunnyvale, CA, USA).

### 2.5. Live/Dead Staining Assay

The AS*vicK* strain exhibited an MBC-B value of 175 μg/mL for DMAHDM. Consequently, the 48-h biofilms treated with DMAHDM (175 μg/mL and 87.5 μg/mL) were stained using a Live/Dead BacLight bacterial viability kit (Eugene, OR, USA). After washing the biofilms three times with PBS, each sample was stained with a mixture of 2.5 μM SYTO 9 and 2.5 μM propidium iodide (PI) for 15 min [43]. Live bacteria emitted green fluorescence upon staining with SYTO 9, while dead bacteria emitted red fluorescence upon staining with PI. The structures of the biofilms were visualized using confocal laser scanning microscopy (CLSM) (Leica, Solms, Germany), and images were captured from three random fields. The Z spacing between the captured layers was set to 2 μm. Three-dimensional reconstruction was performed using the Leica Application Suite X (LAS X) software 3.7.4.23463. The excitation wavelengths for SYTO 9 and PI were 485 nm and 535 nm, respectively, with emission wavelengths of 498 nm and 617 nm, respectively.

### 2.6. Biofilm Viability Using the MTT Assay

The MTT (3-(4,5-dimethyl-thiazol-2-yl)-2,5-diphenyltetrazolium bromide) assay (VWR Chemicals, Solon, OH, USA) is used to investigate the bacterial viability and reproductive capacity of biofilms [24]. After washing the biofilms three times with PBS, 1 mL of MTT dye solution (0.5 mg/mL MTT in PBS) was added to each well and incubated at 37 °C with 90% N_2_ and 10% CO_2_ for 1 h. After removing the MTT solution, 1 mL of dimethyl sulfoxide (DMSO) was added and incubated for 20 min at room temperature to dissolve the formazan crystals. Following thorough mixing via pipetting, the optical density (OD) of the DMSO solution was measured at 570 nm using a microplate reader.

### 2.7. Colony-Forming Unit (CFU) Counts

After washing twice with PBS, the 48-h biofilms were harvested in 1 mL of BHI by scraping and agitation to ensure thorough mixing [44]. The suspended biofilms were then subjected to 10-fold serial dilutions. For each dilution, 10 µL of the solution was placed onto a BHI agar plate in triplicate. The agar plates were incubated at 37 °C with 90% N_2_ and 10% CO_2_ for 48 h, after which the CFU counts were determined by counting the number of colonies.

### 2.8. Crystal Violet (CV) Staining

The UA159 and AS*vicK* biofilm biomass, with or without DMAHDM, was investigated using the crystal violet (CV) assay (Mengbio, Shanghai, China) following a previous study [45]. After 48-h incubation, the biofilms were gently washed three times with PBS and then stained with 0.1% (*w*/*v*) CV, mixed on an orbital shaker for 15 min at 100 rpm and room temperature. After removing the CV stain by washing three times with PBS, 33% acetic acid was added to the biofilms on the orbital shaker for 15 min at 100 rpm. The biofilm mass was quantified by measuring the OD at 595 nm.

### 2.9. Lactic Acid Secretion

The 48-h biofilms were washed twice with PBS, then immersed in 1.5 mL buffered peptone water (BPW) containing 0.2% sucrose and incubated at 37 °C with 90% N_2_ and 10% CO_2_ for 3 h [46]. BPW is composed of 10.0 g/L peptone, 5.0 g/L sodium chloride (NaCl), 3.5 g/L disodium phosphate (Na_2_HPO_4_), and 1.5 g/L monopotassium phosphate (KH_2_PO_4_). Following centrifugation, the supernatant was collected and mixed with reagents from the lactic acid assay kit (Jiancheng Biotechnology, Nanjing, Jiangsu, China). Each tube contained 20 μL of sample or standard liquid, 1 mL of enzyme working solution, and 200 μL of color developer. After thorough mixing, the samples were incubated at 37 °C in a water bath for 10 min. Subsequently, 2 mL of termination solution was added to each sample, mixed thoroughly, and the OD at 530 nm was recorded. The concentration of lactic acid was calculated using the formula: lactic acid (mmol/L) = (A_test_ − A_blank_)/(A_standard_ − A_blank_) × C_standard_ × N, where A represents the OD_530nm_ values, C_standard_ refers to the standard concentration (3 mmol/L), and N denotes the dilution factor.

### 2.10. Biofilm Imaging

The biofilms were visualized using scanning electron microscopy (SEM) (Hillsboro, OR, USA). The 48-h biofilms were gently washed twice with PBS, then fixed in a 2.5% glutaraldehyde solution for 4 h. Glutaraldehyde was dissolved in PBS. After two additional washes with PBS, the specimens were dehydrated in a series of ethanol concentrations (30%, 50%, 75%, 85%, 95%, and 99%) and then sputter-coated with gold [47]. The biofilm specimens were examined at magnifications of 5000× and 20,000×.

To investigate the production and distribution of EPS within the biofilms, a combination of fluorescent dyes was used. A 1 μM solution of Alexa Fluor 647-labeled dextran conjugate (molecular weight 10,000 Da; Life Tech, Carlsbad, CA, USA) was added to the culture medium at the start of biofilm formation to stain the EPS matrix. The bacterial cells in the 48-h *S. mutans* biofilms were labeled with 2.5 µM SYTO 9 (Life Tech, Carlsbad, CA, USA) for 15 min [48]. Biofilms were cultured on coverslips and examined using CLSM with excitation wavelengths of 640 nm for Alexa Fluor 647 and 488 nm for SYTO 9 [48]. The EPS-to-microbe volume ratio and biofilm thickness were analyzed using ImageJ 1.8.0 software.

### 2.11. Polysaccharide Synthesis

The anthrone-sulfuric acid colorimetric assay was used to quantify water-insoluble glucans (WIG) and water-soluble glucans (WSG) [35]. Biofilms were harvested by scraping with an equal volume of PBS and then centrifuged at 12,000 rpm for 5 min at 4 °C. WSG was found in the supernatant, while WIG was located in the precipitate. The supernatant was reserved for WSG measurement using the anthrone method. The WIG-containing precipitate was resuspended in 1 M NaOH and incubated in a constant-temperature water bath at 37 °C for 3 h. The resulting soluble suspension was mixed with anthrone reagent (1 mg/mL in concentrated sulfuric acid) in a 1:3 ratio. The mixture was heated at 95 °C for 10 min, then cooled on ice. Following this, 200 µL of the solution were transferred to a 96-well plate, and the OD at 620 nm was measured using a microplate reader [31]. Standard curves correlating OD_620nm_ with polysaccharide concentrations were constructed using glucose concentrations ranging from 0 to 50 µg/mL.

### 2.12. Quantitative Real-Time-Polymerase-Chain Reaction (qRT-PCR)

qRT-PCR was employed to quantify the expression of selected genes. Briefly, 48-h biofilms were harvested by scraping, and total RNA was extracted using the Trizol method. Reverse transcription was performed using the PrimeScript RT Reagent Kit with gDNA Eraser (TaKaRa, Kyoto, Japan). qRT-PCR was conducted using TB Green™ Premium Ex Taq™ (TaKaRa, Kyoto, Japan) with a two-step PCR protocol. qRT-PCR primers were commercially obtained from Sangon Biotech (Shanghai, China), as detailed in Table 2. Three parallel samples were prepared for each experiment, and gene expression differences between groups were analyzed using the comparative CT (2^−ΔΔCt^) method, with *gyrA* serving as an internal control [47].

### 2.13. Human Tooth Enamel Hardness Measurement

According to the approved plan by the Institutional Review Board, human teeth with sound enamel (caries-free) were obtained from dental school clinics [31]. Enamel slabs with a diameter of 6 mm and a thickness of 2 mm were prepared, as illustrated in Figure 1 using BioRender software (Version 2208) [31]. Acid-resistant nail varnish was applied to cover the entire surface area of the enamel slabs, leaving only the enamel surface exposed. The 42 enamel slabs were randomly allocated into 7 groups, with each group containing 6 slabs. A hardness tester (HVS-10Z, Jingbo Company, Hangzhou, Zhejiang, China) equipped with a Vickers indenter, applying a 50 g load for 20 s, was used to make indentations at the center of the enamel surface [31]. Hardness measurements were recorded both before and after the biofilm acid attacks. Sterile enamel specimens were placed in 12-well plates containing 2 mL of *S. mutans* suspensions at a concentration of 2 × 10^6^ CFU/mL. Each day, the slabs were immersed in 2 mL of BHIS for 4 h at 37 °C with 90% N_2_ and 10% CO_2_. The slabs were then transferred to a new plate with sugar-free BHI medium and incubated for 20 h under the same conditions. To prevent biofilm aging and excessive thickening, which could deprive internal bacteria of nutrients, the biofilms were removed daily using sterile paper. The cleaning process was aided by scraping with pipette tips and washing the plates twice with PBS. Paper towels were placed in individually packaged sterile bags and autoclaved to ensure sterility. A new biofilm was then grown on the plate. This cyclic soaking treatment was repeated daily for 28 days to establish the biofilm model.

### 2.14. Statistical Analysis

Statistical analyses were performed using GraphPad Prism 9.4.1 (GraphPad Software, Inc., San Diego, CA, USA). One-way ANOVA was employed to identify significant effects among the variables. Tukey’s multiple comparison test was used to compare the means of each group. Data are presented as mean ± standard deviation (mean ± SD), and statistical significance was defined as *p* < 0.05.

## 3. Results

### 3.1. Synthesis of DMAHDM and Strain Verification

As shown in Figure 2A, the comparison of the ^1^H-NMR spectra of DMAHDM confirms the identity of the DMAHDM sample. The Fourier Transform Infrared (FT-IR) spectra of DMAHDM displayed characteristic absorption peaks at 1635.08 cm^−1^ (C=C stretching), 1720.52 cm^−1^ (C=O stretching), and 2918.15 cm^−1^ (C-H stretching of saturated hydrocarbons) (Figure 2B). Thus, the identity of the chemical composition of the DMAHDM samples was further confirmed. In addition, UA159 and AS*vicK* strains were confirmed by qRT-PCR analysis (Figure 2C). The AS*vicK* strain inhibits the expression of *vicK*, resulting in a fold change of 0.17.

### 3.2. Susceptibility Testing of DMAHDM

The drug susceptibility of UA159 and AS*vicK* strains to DMAHDM was determined by MIC, MBC-P, and MBC-B. The MIC and MBC-P of the UA159 strain were 7 μg/mL. In contrast, the MIC and MBC-P of the AS*vicK* strain were 3.5 μg/mL (Figure 2D,E). For the AS*vicK* strain, the MBC-B was 175 μg/mL, indicating a 50% reduction compared to the UA159 strain, which exhibited an MBC-B of 350 μg/mL (Figure 2F). Hence, MBC-B (175 μg/mL) and 1/2 MBC-B (87.5 μg/mL) of the AS*vicK* strain were selected as the subsequent experimental concentrations.

Growth curves were analyzed to compare differences between the parent *S. mutans* and the AS*vicK* strain in DMAHDM-containing culture medium. Growth inhibition was observed in planktonic cultures with DMAHDM (Figure 2G,H). Compared to the UA159 strain, the growth of the AS*vicK* strain was more easily inhibited at the same concentration.

### 3.3. Alteration of the Biofilm Formation and Lactic Acid Secretion

Further, the susceptibility of UA159 and AS*vicK* strains to DMAHDM was tested using live/dead staining, CFU counts, and MTT assays. Figure 3A,B shows representative live/dead 3D images and the mean fluorescence intensities in each layer of 48-h biofilms. In these images, living bacteria are stained green, and dead bacteria are stained red. Without DMAHDM, the AS*vicK* strain exhibits a significant reduction in bacterial count compared to the UA159 strain, though both strains predominantly consist of live bacteria. Figure 3C,D shows the mean fluorescence intensities of SYTO 9 and PI, respectively. At 87.5 μg/mL, the SYTO 9/PI ratio was 16.6% for UA159 and 17.3% for AS*vicK* compared to the untreated biofilm. At 175 μg/mL, the SYTO 9/PI ratio was 4.2% for UA159 and 3.0% for AS*vicK* compared to the untreated biofilm (Figure 3E).

In order to determine the effects on biofilm biomass and lactate production capacity, SEM, CV staining, and a lactic acid assay were performed. As shown in Figure 4, under the action of DMAHDM, EPS production, biomass, and lactic acid production were significantly reduced in both AS*vicK* and UA159 biofilms. Figure 4A presents SEM images of typical biofilms formed by the UA159 and AS*vicK* strains. Without DMAHDM, AS*vicK* biofilms exhibited lower density and reduced EPS content compared to UA159 biofilms. When treated with DMAHDM, the biomass of AS*vicK* biofilms was significantly lower than that of UA159 biofilms. This reduction in biomass was more pronounced with increasing DMAHDM concentrations, as indicated by the CV staining results (Figure 4B,C). Figure 4D,E depict the results of the MTT assay and CFU counts of the 48-h biofilms, respectively. Without DMAHDM, AS*vicK* exhibited significantly reduced biofilm metabolic activity and CFU compared to UA159. Moreover, at a concentration of 175 μg/mL, DMAHDM exerted the strongest anti-biofilm effect, reducing CFU by 3 logs for UA159 and 4 logs for AS*vicK* compared to the untreated condition. Figure 4F illustrates the lactic acid production per CFU of the biofilm. Without DMAHDM, the lactic acid production to CFU ratio is the highest, indicating the highest metabolic activity of the biofilm. As the concentration of DMAHDM increases, this ratio gradually decreases, indicating that the presence of DMAHDM leads to reduced metabolic activity of the biofilm.

### 3.4. Inhibition of EPS Production and Biofilms Architecture

The three-dimensional structure of biofilm was visualized using CLSM. Figure 5A,B displays the impact of *S. mutans* AS*vicK* and DMAHDM on EPS production and biofilm architecture. Regardless of DMAHDM presence, the AS*vicK* biofilms exhibited fewer bacteria and less EPS compared to the UA159 biofilms. Additionally, the EPS/microbe volume ratio and biofilm thickness were quantified (Figure 5C,D), revealing a significant decrease with increasing DMAHDM concentration in the AS*vicK* strain.

The EPS of *S. mutans* biofilms primarily consists of glucans, including water insoluble glucans (WIG) and water-soluble glucans (WSG). The anthrone-sulfuric acid colorimetric assay was employed to detect WIG and WSG and to explore the effects of *S. mutans* AS*vicK* and DMAHDM on EPS content. Using glucose as the standard, we generated a calibration curve (Y = 0.004676X + 0.08316) with a coefficient of determination (R^2^) of 0.9939, which facilitated the calculation of polysaccharide concentrations. Figure 6A,B demonstrates that AS*vicK* significantly reduced both WIG and WSG compared to the UA159 strain. DMAHDM also exhibited a dose-dependent effect on reducing WIG and WSG. Due to AS*vicK*’s already reduced EPS levels compared to UA159, DMAHDM had a stronger inhibitory effect on WIG and WSG in UA159. Specifically, at 87.5 μg/mL, WIG was reduced by 57.7% in UA159 and 46.1% in AS*vicK*. At 175 μg/mL, reductions were 74.8% for UA159 and 67.8% for AS*vicK*. For WSG, reductions at 87.5 μg/mL were 48.5% for UA159 and 11.1% for AS*vicK*, and at 175 μg/mL, reductions were 58.2% for UA159 and 13.3% for AS*vicK*. Notably, AS*vicK* with 175 μg/mL DMAHDM reduced both WIG and WSG by over 70% compared to UA159 without DMAHDM.

### 3.5. Regulation of Gene Expression Related to EPS Metabolism

The *vicK*, *gtfB*, *gtfC*, *gtfD*, and *ftf* genes are involved in EPS synthesis. Figure 6C–G illustrates their expression levels. As shown in Figure 6C, DMAHDM inhibits the expression of *vicK* in the UA159 strain, with fold changes of 0.83 at 87.5 μg/mL and 0.39 at 175 μg/mL. Without DMAHDM, the AS*vicK* strain shows reduced expression of *gtfB*, *gtfC*, *gtfD*, and *ftf*, with fold changes of 0.54, 0.75, 0.67, and 0.30, respectively, compared to UA159. At 87.5 μg/mL DMAHDM, UA159 demonstrates significant down-regulation of *gtfB*, *gtfC*, *gtfD*, and *ftf*, with fold changes of 0.76, 0.52, 0.59, and 0.73. Similarly, AS*vicK* shows significant down-regulation of these genes, with fold changes of 0.04, 0.13, 0.42, and 0.53. At 175 μg/mL DMAHDM, both strains exhibit notable down-regulation, with UA159 showing fold changes of 0.51, 0.54, 0.43, and 0.20, and AS*vicK* showing fold changes of 0.02, 0.11, 0.03, and 0.43. The combined effect of *S. mutans* AS*vicK* and 175 μg/mL DMAHDM results in a substantial decrease in the expression of *gtfB*, *gtfC*, *gtfD*, and *ftf*, with fold changes of 0.01, 0.08, 0.02, and 0.13, respectively, compared to UA159 without DMAHDM.

### 3.6. Protection of Human Tooth Enamel Hardness

The hardness of human tooth enamel was measured to study the effect of *S. mutans* AS*vicK* and DMAHDM on inhibiting enamel demineralization. Enamel hardness at the enamel surface was plotted in Figure 7. The sound enamel hardness was measured at 3.91 ± 0.19 GPa. Following 28-day exposure to *S. mutans* biofilm acid attack, enamel hardness decreased across all groups, with the greatest reduction observed in the UA159 strain without DMAHDM (0.64 ± 0.05 GPa), which is just 16.4% of sound enamel hardness. As the DMAHDM concentration increased to 87.5 μg/mL or 175 μg/mL, the erosion-preventing effect was significantly enhanced, with a more pronounced effect observed in AS*vicK* biofilms (Figure 7). At 87.5 μg/mL, enamel hardness was 46.9% for UA159 and 53.6% for AS*vicK* of sound enamel hardness. At 175 μg/mL, it increased to 57.9% and 67.1%, respectively. Without DMAHDM, the enamel hardness with the AS*vicK* strain is 164% of that with the UA159 strain. Notably, the enamel hardness with the AS*vicK* strain combined with 87.5 μg/mL DMAHDM was 320% of that with the UA159 strain without DMAHDM. Moreover, the enamel hardness with the AS*vicK* strain and 175 μg/mL DMAHDM was 2.62 ± 0.24 GPa, 410% of that with the UA159 strain without DMAHDM.

## 4. Discussion

DMAHDM, a quaternary ammonium monomer with a chain length of 16 [25,37,49], demonstrates potent antimicrobial activity against a range of microorganisms, including bacteria, fungi, and certain viruses [22]. Its mechanism of action involves the positively charged quaternary ammonium compounds interacting with the negatively charged cell membranes, causing an electrical imbalance. This imbalance leads to increased osmotic pressure, resulting in microbial cell swelling and eventual rupture with cytoplasmic leakage [50]. Although DMAHDM has strong antimicrobial properties, it does not affect all normal oral microbiota. Our previous research showed that DMAHDM can increase non-cariogenic species, such as *Streptococcus sanguinis* and *Streptococcus gordonii*, in a multi-species biofilm model [3,14,51]. The effectiveness of DMAHDM can be diminished when *S. mutans* forms biofilms [37]. *S. mutans*, in particular, can develop robust, mature, three-dimensional biofilms that protect bacteria from environmental stressors and antimicrobial agents [52]. In our study, the MIC, MBC-P, and MBC-B of the AS*vicK* strain were reduced by 50% compared to the UA159 strain, indicating a two-fold increase in susceptibility to DMAHDM. Additionally, the AS*vicK* strains showed a significant reduction in extracellular matrix production and disruption of the dense biofilm structure. The reduced biofilm biomass allowed greater exposure of bacteria within the biofilm, enhancing the “contact inhibition” effect of DMAHDM and increasing bacterial susceptibility to the compound.

In this study, we demonstrated that DMAHDM effectively inhibits biofilms formed by both UA159 and AS*vicK* strains. This inhibition is associated with a reduction in *vicK* gene expression, which is further decreased when using AS*vicK*. The combination of *S. mutans* AS*vicK* and DMAHDM significantly suppresses EPS production and disrupts the three-dimensional structure of the biofilms. To understand this effect, we investigated the underlying mechanism of EPS synthesis inhibition by AS*vicK* and DMAHDM, focusing on their regulatory effects on EPS metabolism-related gene expression, including *gtfB*/*C*/*D* and *ftf*. Glucosyltransferases (Gtfs) and fructosyltransferases (Ftfs) are critical for EPS synthesis [53]. Gtfs function as transglycosylases, catalyzing the conversion of glucose into glucans, which are essential for EPS formation through glucosidic bonds. In contrast, Ftfs catalyze the formation of fructosan from fructose, serving as extracellular storage compounds [54]. Previous studies have shown that inhibiting the *vicK* gene down-regulates the expression of *gtfB*/*C*/*D* and *ftf* [55,56], which aligns with our results that AS*vicK* down-regulates these genes. The synergistic effect of *S. mutans* AS*vicK* and DMAHDM enhances gene downregulation. Together, *S. mutans* AS*vicK* and DMAHDM inhibit EPS synthesis. This reduces the protective barrier function of EPS against antimicrobial agents, thereby diminishing the cariogenic virulence of *S. mutans*.

In an in vitro caries model, we confirmed the dual efficacy of combining *S. mutans* AS*vicK* with DMAHDM in suppressing enamel demineralization. After 28-day exposure to *S. mutans* biofilm acid attack, enamel hardness was significantly higher in samples treated with DMAHDM. Samples with AS*vicK* strain biofilms exhibited increased hardness compared to those with UA159 strain biofilms. Notably, DMAHDM provided a significantly greater protective effect than AS*vicK* alone. Furthermore, the suppression of enamel demineralization became more pronounced with increasing concentrations of DMAHDM, with the combination of *S. mutans* AS*vicK* and 175 μg/mL DMAHDM showing the most significant effect.

Our study confirmed the dual effect of *S. mutans* AS*vicK* combined with DMAHDM in inhibiting both biofilm formation and enamel demineralization. Additionally, it confirmed the role of AS*vicK* in enhancing antimicrobial susceptibility to DMAHDM. Nonetheless, two limitations merit consideration. Firstly, our investigation solely focused on the effects of *S. mutans* AS*vicK* and DMAHDM in a single-species biofilm. However, oral dental plaque comprises a diverse array of bacteria, and its properties arise from the interaction of multiple microorganisms. Secondly, the in vitro cariogenic model we employed utilized human enamel slabs immersed in BHI medium rather than replicating the conditions of saliva-based human or animal oral environments. In the oral environment, biofilms are subject to toothbrushing and mouthwash treatments, which reduce acid production. In contrast, biofilms grown in BHI medium continue to produce acid unabated, resulting in accelerated enamel caries development [57]. This is the primary reason why the enamel hardness of the best-performing biofilm was significantly lower than that of sound enamel. Therefore, future studies should investigate how AS*vicK* and DMAHDM affect a more complex microbial community.

Currently, enhancing the susceptibility of anti-caries drugs, inhibiting biofilm formation, and protecting tooth structure are crucial factors in evaluating the clinical viability of new anti-caries approaches [32]. Recent research into regulatory RNAs and their role in antibiotic response and resistance across various bacterial pathogens highlights their importance [58]. AS*vicK*, an antisense RNA that complements *vicK* mRNA, could serve as an effective biomarker for bacterial infections and as a novel therapeutic agent in drug design. However, the high bacterial turnover rate in the oral cavity might impact the effectiveness of antisense RNA. Therefore, an appropriate delivery vector is essential to protect antisense RNA from nuclease degradation and ensure effective gene delivery to *S. mutans*. Our study confirmed that DMAHDM has favorable biocompatibility and strong antibacterial activity. Future research could explore additional clinical applications for DMAHDM-based anti-caries products, such as mouthwashes and toothpastes. The novel strategy of combining bacterial gene-modified strains with antibacterial monomers holds promise for combating dental caries. This approach also helps preserve the diversity of normal oral flora. Nonetheless, realizing these prospects will require further research and development to fully explore and implement these innovative approaches.

## 5. Conclusions

Our study’s results demonstrated that: (1) DMAHDM effectively inhibited biofilm formation and EPS synthesis; (2) *S. mutans* AS*vicK* increased the susceptibility of *S. mutans* to DMAHDM; and (3) the combination of *S. mutans* AS*vicK* and DMAHDM showed enhanced effectiveness in inhibiting biofilm formation and reducing enamel demineralization. Overall, combining bacterial gene-modified strains with the antibacterial monomer (AS*vicK* + DMAHDM) shows significant potential for anti-caries applications.

## Figures and Tables

**Figure 1 pathogens-13-00707-f001:**
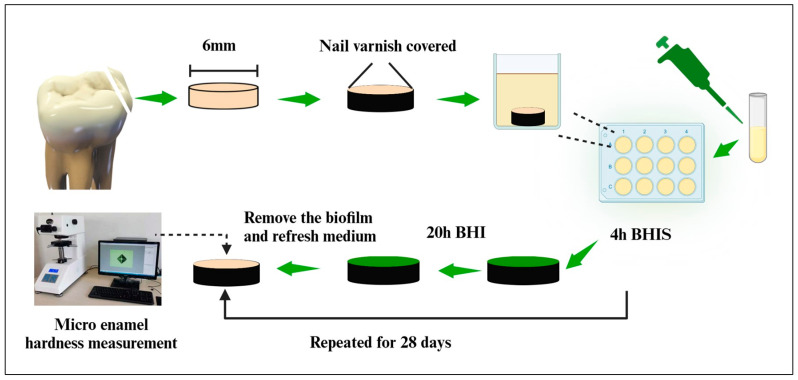
Schematic diagram illustrating the experimental design process for suppressing tooth enamel demineralization in vitro.

**Figure 2 pathogens-13-00707-f002:**
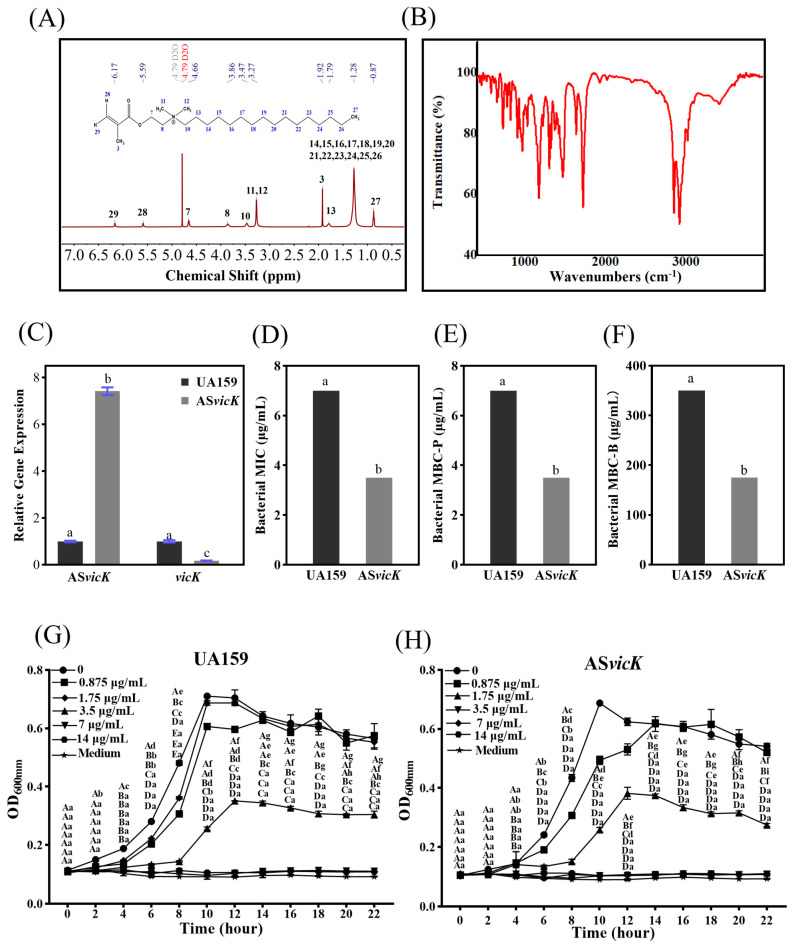
The chemical structure of Dimethylaminohexadecyl methacrylate (DMAHDM) and the drug susceptibility of UA159 and antisense *vicK* RNA (AS*vicK*) strains are presented. (**A**) Hydrogen-1 nuclear magnetic resonance (^1^H-NMR) spectra; (**B**) Fourier-transform infrared spectroscopy (FTIR) spectra; (**C**) quantitative real-time PCR (qRT-PCR) verification (mean ± SD; *n* = 3); (**D**–**F**) minimum inhibitory concentration (MIC), minimum bactericidal concentration of planktonic bacteria (MBC-P), and minimum bactericidal concentration of biofilms (MBC-B) (mean ± SD; *n* = 3); Growth curves of (**G**) the UA159 and (**H**) AS*vicK* strains treated with different mass fractions of DMAHDM for 22 h (mean ± SD; *n* = 3). Different letters represent significantly different values (*p* < 0.05).

**Figure 3 pathogens-13-00707-f003:**
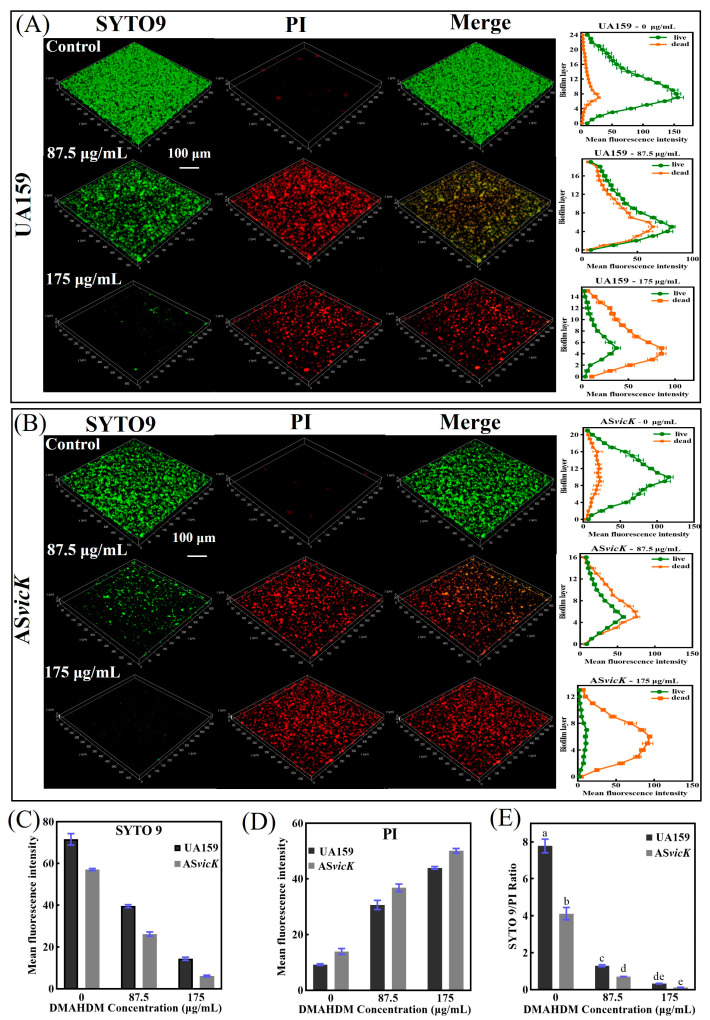
Representative live/dead staining images of preformed biofilms treatment of UA159 and AS*vicK* strains: (**A**) Live/Dead staining images and the mean fluorescence intensities for live and dead bacteria in each layer of the UA159 strain (mean ± SD; *n* = 3); (**B**) Live/Dead staining images and the mean fluorescence intensities for live and dead bacteria in each layer of the AS*vicK* strain (mean ± SD; *n* = 3); (**C**) Mean fluorescence intensity of SYTO 9 (mean ± SD; *n* = 3); (**D**) Mean fluorescence intensity of PI (mean ± SD; *n* = 3); (**E**) SYTO 9/PI Ratio (mean ± SD; *n* = 3); Different letters represent significantly different values (*p* < 0.05).

**Figure 4 pathogens-13-00707-f004:**
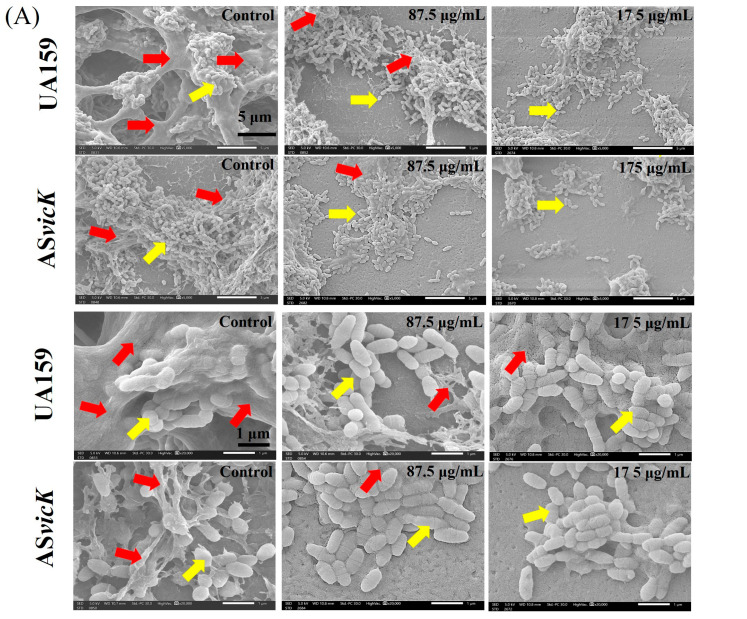
Representative SEM images, activity, and virulence factors of preformed UA159 and AS*vicK* biofilms treated with DMAHDM: (**A**) SEM images, with the yellow arrows pointing to bacteria while the red arrows point to EPS; (**B**) Crystal violet staining images; (**C**) Biofilm biomass (mean ± SD; *n* = 6); (**D**) MTT metabolic activity (mean ± SD; *n* = 6); (**E**) Colony-forming units (CFUs) (mean ± SD; *n* = 3). (**F**) Lactic Acid Production/CFU (mean ± SD; *n* = 3). Different letters represent significantly different values (*p* < 0.05).

**Figure 5 pathogens-13-00707-f005:**
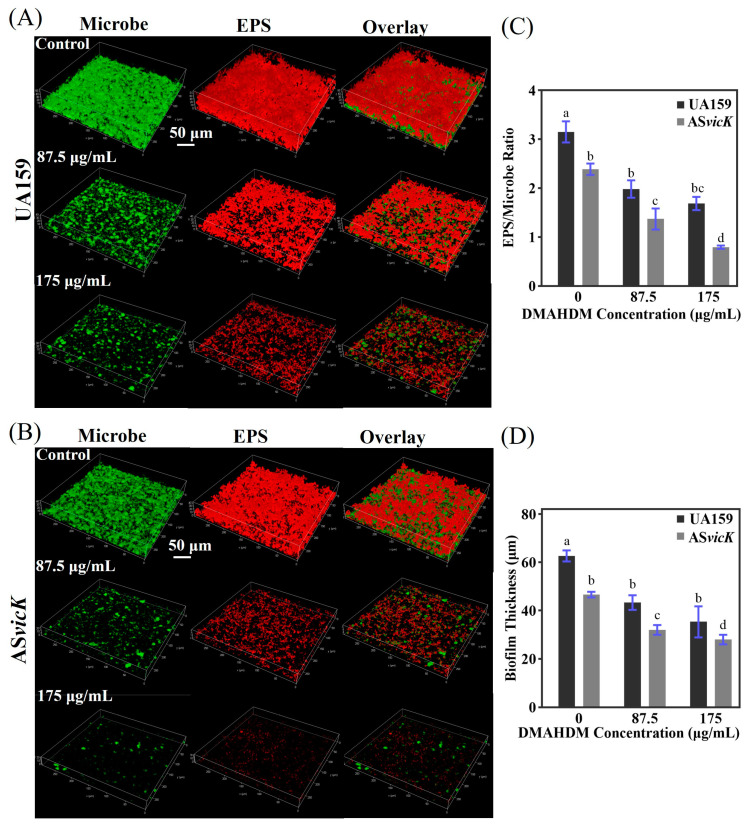
Representative CLSM images of preformed UA159 and AS*vicK* biofilms treated with DMAHDM. The EPS was labeled with Alexa Fluor 647, and the bacteria were labeled with SYTO 9. (**A**) CLSM of the UA159 biofilms; (**B**) CLSM of the AS*vicK* biofilms; (**C**) EPS/Microbe volume ratio of UA159 and AS*vicK* (mean ± SD; *n* = 3); (**D**) Biofilm thickness of UA159 and AS*vicK* (mean ± SD; *n* = 3). Different letters represent significantly different values (*p* < 0.05).

**Figure 6 pathogens-13-00707-f006:**
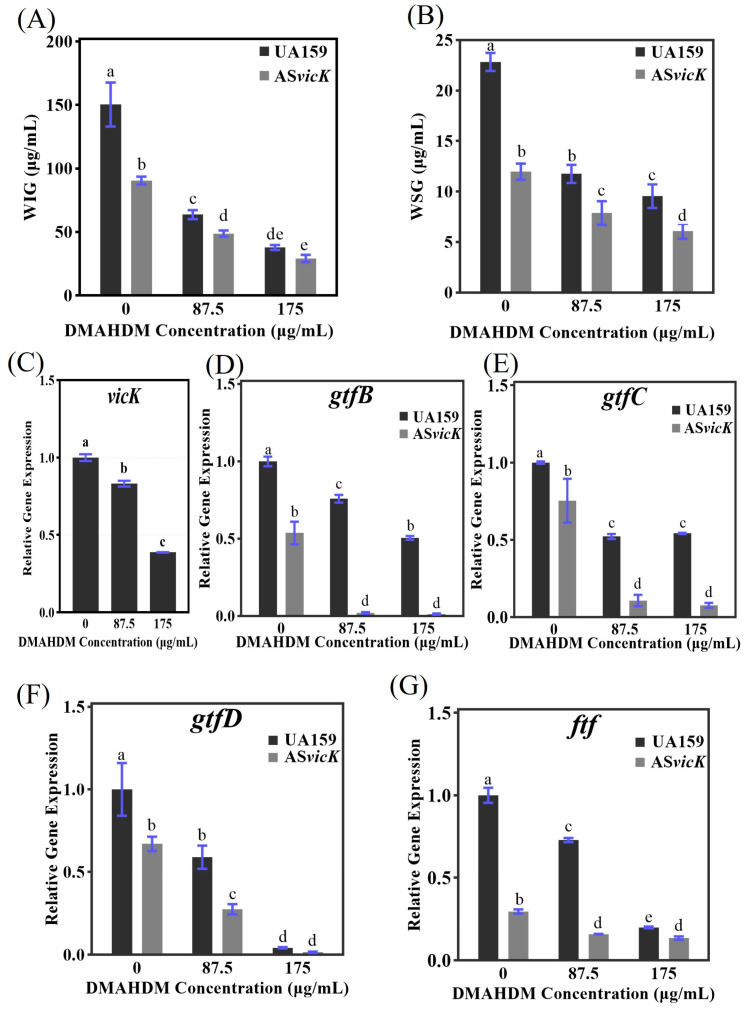
Polysaccharide amounts and gene expression related to EPS metabolism were analyzed. The biofilm was incubated with BHI containing 1% sucrose for 24 h, followed by incubation with BHI containing DMAHDM for an additional 24 h. (**A**) Production of WIG (mean ± SD; *n* = 6); (**B**) Production of WSG (mean ± SD; *n* = 6); (**C**–**G**) Gene expression of *vicK*, *gtfB*/*C*/*D* and *ftf* via qRT-PCR (mean ± SD; *n* = 3). Different letters represent significantly different values (*p* < 0.05).

**Figure 7 pathogens-13-00707-f007:**
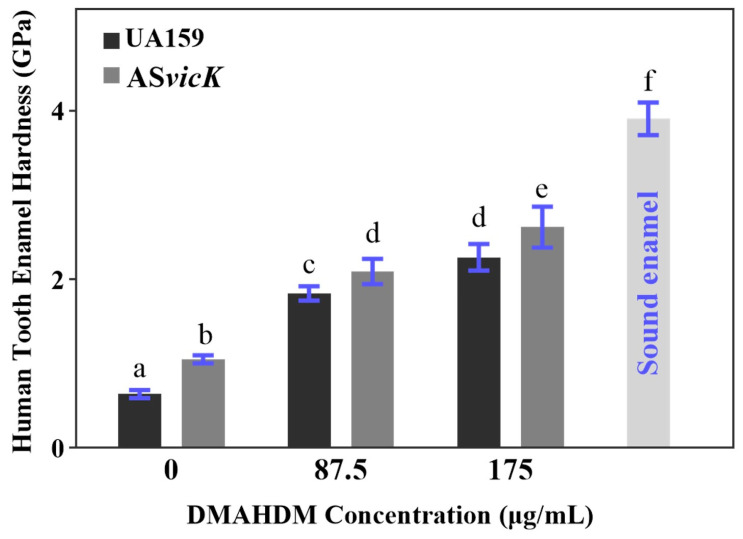
Hardness of human tooth enamel at the surface after a 28-day acid attack by preformed *S. mutans* biofilms. The biofilms were treated with 87.5 and 175 μg/mL DMAHDM, respectively. The hardness of sound enamel and demineralized enamel was also measured as comparative controls. The hardness test was conducted at the center of the enamel slab. (mean ± SD, *n* = 6). Different letters represent significantly different values (*p* < 0.05).

**Table 1 pathogens-13-00707-t001:** Bacterial strains and plasmids used in this study.

Strains or Plasmids	Description	Source or Reference
Parent *S. mutans*	*Streptococcus mutans* UA159	^a^ATCC 7000610
AS*vicK S. mutans*	AS*vicK* over expression strain	[13]

^a^ATCC: American Type Culture Collection.

## Data Availability

Data obtained in this study will be available upon reasonable request from the corresponding authors.

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
