# Peer review of "S. mutans Antisense vicK RNA Over-Expression Plus Antibacterial Dimethylaminohexadecyl Methacrylate Suppresses Oral Biofilms and Protects Enamel Hardness in Extracted Human Teeth"

_pathogens, 2024, doi:10.3390/pathogens13080707_

Round 1

Reviewer 1 Report

Comments and Suggestions for Authors

In this manuscript, the authors describe studies aimed to investigate the ability of antisense S. mutans vicK mRNA (ASvicK), combined with the quaternary ammonium antibacterial compound DMAHDM to influence biofilm formation in an in vitro biofilm model.  Enamel slabs prepared from caries-free human teeth were used as substrate for biofilm formation.  The enamel hardness was measured after 28 days of exposure to S. mutans culture.

When testing by MIC and MBC in both planktonic and biofilm cultures, the ASvicK strain showed greater susceptibility (by 2-fold) to DMAHDM than did the parent strain UA159.  Biofilm formation and extracellular polysaccharide production were notably compromised in the ASvicK strain.

This manuscript is logical, well-written, and relevant.  I feel that the manuscript would be vastly improved if two major aspects were addressed:

1.       The Introduction and reasoning for performing each experiment must be expanded.  In the Results section, the authors hastily jump straight into the outcome without even describing the reason that each experiment was performed.

2.       As I see it, ASvicK alone and DMAHDM treatment alone each cause notable disruption in biofilm formation, which directly relates to the other described phenotypes.  The authors have already explored each of these aspects on their own.  The authors should provide a strong case for combining these two aspects.  In the manuscript’s current form, I am not convinced that the combination of ASvicK and DMAHDM provides improved results as compared to either condition on its own.  The authors say it themselves in line 365: “ASvicK and DMAHDM, either alone or in combination, exerted significant suppressive effects…”.   The authors need to convey why it would be worthwhile to combine these.  As I see it, disrupting vicK expression results in a huge defect in biofilm formation.  While it is true that the combination of the two treatments is helpful in the enamel hardness test, the MTT assay results (Fig. 3B) make one wonder if the DMAHDM treatment would have devastating results on a patient’s normal flora.  I feel that in an attempt to highlight DMAHDM, the authors dance around the drastic effect that ASvicK has on S. mutans biofilm formation (for example, the different in untreated strains is not acknowledged in the text describing Figs. 3 and 4).

Additional points:

1.       Line 78:  Why add the sentence about the literature search for the impacr to ASvicK on the susceptibility of S. mutans to DMAHDM?  This seems strangely specific. 

2.       Line 93:  the word “teeth” is missing at the end of the sentence.

3.       Lines 104, 106:  Placeholders (***) are still filling in for details.

4.       Line 107:  Please provide more detail about the ASvicK over-expression plasmid

5.       Lines 167:  Please change the language to say “Subsequently, the suspended biofilms were subjected to 10-fold serial dilutions which were spread onto BHI agar plates.”

6.       The role of lactic acid is not mentioned in the Introduction

7.       Please provides guidelines as to what is expected (or not expected) in Figs 2A and 2B.

8.       Fig. 2D is unnecessary.  Please remove.

9.       Were the differences in Fig. 2E, F, G not statistically significant?

10.   It would be nice if the Results section included a bit more narrative.  In its current state, the authors abruptly jump straight to the results of each experiment without describing the reasoning for why it was performed.

11.   The authors do not discuss the significance of performing the MTT assay. 

12.   Fig. 3B:  The MTT assay shows that DMAHDM has major metabolic consequences that are likely to be equally suffered by other oral bacteria that would be exposed to a treatment regimen.  The authors must address this.

13.   VicK is known to play a role in biofilm formation.  It is not surprising to see that the ASvicK biofilms are patchier than those formed by UA159, even in the absence of DMAHDM. This should be acknowledged (Note the difference in the first two bars of the graph in Fig. 3C).

14.   Lines 279-280:  The authors claim that “The biomass of ASvicK biofilms combined with DMAHDM was significantly lower than that of UA159 biofilms in the no-drug group.”  It is not a fair comparison to compare treated ASvicK to untreated UA159.  The lack of vicK expression clearly has a negative impact upon biofilm formation that overrides any impact of DMAHDM described in Fig. 4.

15.   Why aren’t Fig. 5C and 5D combined in the manner that Fig. 5E is shown?

16.   The genes chosen as related to EPS metabolism are not described in terms of their roles and contribution to biofilm formation.

17.   Lines 343-348, 360-364 would be more effective in the Introduction.

Author Response

Dear Reviewer:

I hope you are doing well. Please see the attachment.

Reviewer 2 Report

Comments and Suggestions for Authors

Comments to pathogens-3063996

The manuscript studied how anti-sense to vicK increases the susceptibility of the cariogenic Streptococcus mutans to dimethylaminohexadecyl methacrylate (DMAHDM).   As a mechanism of action, this manuscript provides novel data. Concerning its clinical applicability, it is questionable how anti-sense RNA can be used to treat bacteria in the oral cavity where the bacterial turnover is high. Another issue that needs to be addressed is the cytotoxicity of DMAHDM.

There are several flaws in the paper as stated below. The reliability of the images should be confirmed. Please address all of the issues mentioned below by doing revision in the text and figures.

·         The following references should be added to the manuscript: DOI: 10.1016/j.jdsr.2021.03.003; DOI: 10.3390/ma14071688; DOI: 10.1016/j.jdsr.2021.03.003

  • In the graphic abstract: No need to add numbers in the formula. The authors should prepare the formula by themselves. It is misleading to make an inhibitory sign between ASvicK to gtfB, gtfC and gtfD. The anti-sense is acting on vicK which in turn affects the expression of these genes. The abstract should stress that DMAHDM is used as dental material for tooth, and how " ASvicK enhances the susceptibility of S. mutans to DMAHDM , as stated in the abstract.
  • Line 47: Syntactic error: Please correct to "exopolysaccharide" without "s".
  • Line 63: You can delete: "And then".
  • Line 66: What do you mean by: "non-releasing". This should be better explained.
  • Lines 70 and 71: There is an extra hyphen in each line that should be removed.
  • Lines 88-89: Delete "the" before "biofilm formation".
  • I am not sure if there is any need to add the "hypotheses" in lines 89-93, as you described the objectives. Instead of the sentences in lines 89-93 that do not contribute much to the manuscript, it would rather be preferable to describe the reasoning behind the combination, and add to the Introduction which genes are regulated by VicRK.
  • Please devote a paragraph in the Introduction to describe the VicRK system.
  • Section 2.1: What was the purity of DMAHDM? And which impurities were retained?
  • Line 104 has three asterisks – please state the name. Also state approval for what.
  • Line 106 has three asterisks – please write the source.
  • Line 107: The name of the plasmid used should be stated.
  • Line 108: Add in brackets (Table 2) after stating " conducted using quantitative real-time PCR (qRT-PCR)". Please state which antibiotics were used for confirming the strains. Also add the concentrations of the antibiotics.
  • Line 109: Delete "was". State which "confirmation" experiments.
  • Line 110: small p letter in "parental"
  • The city and state of any sources should be mentioned.
  • Line 112: Please add a space between "1" and "mg".
  • Line 113: Was the CFU determined by counting or by optical density?
  • In Line 115: I think it should be mentioned that "the biofilms formed on the well surface were washed".
  • Line 116: How was DMAHDM dissolved? Please state the concentrations used. Did it form suspension? Explain better what you mean by "mass fraction".
  • Line 126: Correct to: "overnight" without hyphen.
  • Line 126: What was the volume of bacteria used? What was the final incubation volume.
  • Line 129: The optical density should be read in a plate reader.
  • A kinetic study to show the changes in OD over time should be presented.
  • Line 135: A double-sword: First written "anaerobic" then 10% CO2". Which of the statements is right?
  • Line 142: How did you prepare single bacteria suspension from biofilms?
  • Line 147: The MBC-B is quite high. What was the MBC-B of the wt strain?
  • Line 151: Correct to SYTO 9.
  • Line 154: The excitation/emission wavelengths and the size of the steps between each layer should be mentioned.
  • Line 160: I think it is worth mentioning that the MTT solution was removed before adding DMSO.
  • Line 173: correct to: "staining"
  • Line 173: correct to: "and mixed on an orbital shaker".
  • Line 174: Again: Before adding acetic acid, the CV stain should be removed by several washes. This has to be stated.
  • Line 188: The following sentence should be rephrased: "The inhibitory effects of biofilms". It says that the biofilms have an inhibitory effect. You can simply state: "The biofilms were visualized by".
  • Line 190: The solvent of the glutaraldehyde solution should be stated.
  • Line 190: The wash solution after fixation should be stated.
  • Line 193: Wat about higher magnifications?
  • Line 194: correct to "48 h-".
  • Line 196:  AlexaFluor647 is the fluorochrome. The conjugated substance, I believe, is Dextran 10,000. (It should be added during incubation in order to detect all EPS in each biofilm layer). You are sure that it is 1 μM and not 1 μg/mL?
  • Lines 196 and 198: Correct to SYTO 9.
  • Lines 199-201: The steps should be stated, and the software used to make the 3D structures.
  • Line 198: It is not "at a range of". These are the excitation laser wavelengths. Please correct.
  • Line 209: Please state the composition of the anthrone reagent.
  • Line 221: Correct to CT.
  • Line 224: Delete the three asterisks.
  • Line 233: Please state for how many days this was done.
  • Figure 1: How did you ensure that all biofilms were removed?
  • Line 237: Which kind of paper was used, and how did you keep it sterile?
  • Section 2.13: Please state the name of the statistical test.
  • Line 249: Please write "by" instead of "through".
  • Figure 2: A. There are two peaks not labelled with a number. What are they? And, fl should be defined.
  • Figure 2B: A space should be added in the title of the Y-axis before the first bracket. And the title of the X-axis should be close to the axis an not outside the box!
  • Figure 2C: Standard deviation should be added to UA159 columns. The number of replicates should be stated.
  • Figure 2D: The antibiotic in the agar plate should be stated.
  • Figure E-G: std is lacking. State in legend how many repeats were done. Add statistics.
  • Line 258: Add the MIC of ASvicK strain.
  • Lines 164-165: If both strains show predominantly live bacteria, how could it be that they "exhibited different phenotypes"?
  • Line 268: Instead of "groups", I would state "biofilms".
  • Figure 3A: The average fluorescence intensities should be calculated for each layer, and presented in a graph. The PI/SYTO 9 ratio should be calculated and presented. This is in light of the weaker biofilms formed by ASvicK. In A: Are these merged images? Strange that there is no SYTO 9 staining at all in the high concentration, as SYTO 9 stains both live and dead bacteria, while PI only the dead bacteria. The single images of each fluorescence should also be shown.
  • According to figure 3: The metabolic activity is affected more than the CFU – suggesting an anti-metabolic effect. Figure 3A is in contrast to Figure 3C – please check the correctness of Figure 3A.
  • Line 277: Correct to "presents".
  • The following sentence has to be rephrased: "The biomass of ASvicK biofilms combined with DMAHDM was significantly lower than that of UA159 biofilms in the no-drug group". In the beginning of the sentence the combined treatment with DMAHDM is stated, while at the end it says: " the no-drug group". What do you mean?
  • Figure 4: I have never seen such a separation between bacteria and EPS. Usually the bacteria are enwrapped in the EPS. Did you use different conditions to get such EPS? There is an increase in EPS in ASvicK with 87.5 μg/mL DMAHDM. So red arrows should be added to this subfigure.
  • The legend of Figure 4 lacks explanation that these biofilms were treated with DMAHDM.
  • The strongly reduced lactic acid production by DMAHDM is an indication for reduced metabolic activity.
  • Does Figure 5 go along with Figure 4A with respect to EPS?
  • Similarly, does Figure 5 go along with Figure 3A? At least here in Figure 5 you can see the SYTO 9 staining that was not seen in Figure 3A. The legend to Figure 5 should provide more detailed information. E.g., the type of staining.
  • Since Figure 5E is a combination of the two strains, Figure C and D should similarly be combined.
  • Line 306: correct to: "illustrate" (present tense).
  • Looking at Figure 6A: The anti-WIG and anti-WGS effect of DMAHDM was stronger for wt than ASvicK.  The percentage reduction should be mentioned in the text. This can be due to the already reduced EPS in ASvicK in comparison to wt.
  • In line 310 – I understand the thinking of the authors: Everything has been compared to untreated UA159 or vehicle-treated??). But you should compare treated versus untreated sample of the same strain. Please correct the text accordingly everywhere.
  • Figure 6C-F should also have a std on untreated UA159 (a comparison between the different untreated UA159 samples).
  • How does DMAHDM affect vicK expression? Also, was it the basic vicK level in UA157? And which factors affect its expression. I mean, that if you induce vicK expression in the bacteria, would it affect the anti-bacterial activity of DMAHDM?
  • On the other hand, the inhibition of gtfB, gtfC and gtfD genes by DMAHDM was stronger on ASvicK than wt, suggesting that the gene repression by DMAHDM is vicK-independent, and antagonized by vicK. Could it be? However, ftf was mainly repressed by DMAHDM in wt, suggesting that its repression depends on vicK.
  • Line 316: Instead of "were depicted" please write "are shown".
  • According to Figure 6C-F – ftf is the most affected by ASvicK – ftf should be added to the graphic abstract.
  • Figure 6: Was the gene expression done in the presence or absence of sucrose? Please indicate the time of incubation with DMAHDM.
  • Section 3.5. The authors only discuss changes in gene expression in ASvicK, but they should also do it for wt treated with DMAHDM. Again, the repression should be calculated for treated samples versus control samples (vehicle treated) of the same strain. The text needs to be corrected.
  • Legend to Figure 7 should prove a better description. If 175 μg/mL DMAHDM totally prevent lactic acid production (Figure 4D), and most of the bacteria are dead with this concentration (Figure 3A), then how can you explain that DMAHDM at this concentration only partly rescued the enamel?
  • Line 340: There is a red point.
  • In discussion: Please exchange the word "group" with another proper word, e.g., "biofilm".
  • Line 358: Was DMAHDM in solution or the surface was coated? What do you mean by "contact" inhibition? This needs to be better described.
  • Lines 370-372: Gtf and ftf are transglycosylases, so please correct the sentence.
  • Lines 378-393: There was only small changes between wt and ASvicK with regards to tooth demineralization, such that the reduced demineralization in the presence of DMAHDM is a direct effect of DMAHDM almost independent of ASvicK. Thus, the text should be rephrased accordingly. The contact-inhibition in line 390 should be explained.

Comments on the Quality of English Language

The English is in general fine, with some scattered mistakes. So Moderate English editing is required.

Author Response

(The authors gave the same response as above.)

Round 2

Reviewer 1 Report

Comments and Suggestions for Authors

I feel that the modifications have improved the manuscript considerably.  I really appreciate the the authors took the reviewers' comments to heart.

I have no additional suggestions; the revised manuscript is acceptable.

Author Response

(The authors gave the same response as above.)

Reviewer 2 Report

Comments and Suggestions for Authors

Comments to revised version:

Still many sentences that need English editing. Remember to have a space before the reference brackets.

I would delete from the Title "Novel strategy of".

Try to remove unnecessary wordings in the abstract, and make it concise.

It should be stressed in the abstract that you compared the susceptibility of two S. mutans strains – parental and an ASvicK overexpressing strain to DMAHDM. ASvicK was not provided as a therapy method where it is added to the medium.  Thus, the title and the abstract should be corrected accordingly to provide exact information.

Line 18: Delete "the" before "cariogenic"

Since Dimethylaminohexadecyl methacrylate is not a novel compound, I would delete "novel" in the abstract line 19.

Line 20: There is no need to write "that has been synthesized", which can be deleted.

Lines  20-23: The sentence is long and has an repetition element.  When you are studying the impact of antisense vicK RNA it is obvious that it has been combined with DMAHDM, so the second part of the sentence can be deleted. (It says the same). This sentence should therefore be rephrased. Prior to the sentence it should be stated that DMAHDM has anti-bacterial activity.

Line 26: The word "evaluated" is not scientific. You can say "studied" or "investigated".

The same in Iine 28: You can rephrase the following text: " These evaluations were conducted using". You need to state which combinations were studied.

Lines 30-31: State the MIC values and the comparison. In the current text it is not obvious what is compared to what. This sentence should also appear before the cariogenic model.

Line 34: How did you come to "4.1-fold" higher enamel "hardness". This has to be defined. Actually, the assay shows prevention of demineralization, such that the numbers should be percentage demineralization. The fold change here is not accurate, since you compared it to demineralized tooth and not the intact tooth.

How does DMAHDM cause anti-bacterial activity when it is a polymer and not soluble? (It says in the text that is it "non-released", but later that it was dissolved – this is quite confusing).

Line 65: " thereby bolstering bacterial tolerance" the sentence lacks information about tolerance to what?

Please add a space between "RNA" and "in".

Lines 71-72: Describe exactly the post-transcriptional regulatory effects on biofilm.

The sentence in lines 94-87 should have references. The abbreviations MPC, nCaF2 and NACP should be defined.

Line 89: Concerning the sentence: " DMAHDM can reduce S. mutans" the sentence is incomplete. Could it be that you meant: "reduce the amount of S. mutans" also state in which system (oral cavity/tooth surface).

The sentence in lines 99-100 is incomplete. Please correct.

The same problem as in abstract, reappears in lines 102 and 103 from a logical point of view. What is the difference between points 1 and 2? Both use combined treatment.

Again, it should be stated that you used a strain carrying the ASvicK, and this is not the same as treating parental bacteria with this anti-sense RNA. So, the text should be rephrased.

Line 116 and line 292: correct to: "Strain verification" (without s ).

Lines126 and 131: Check the correctness of the spectinomycin concentration. Is it really 1 mg/ml.

Line 135: If DMAHDM is dissolved in water, how can you say that it is not released? Clarify this in the text.

Line 167: You can delete "And".

Line 186: Correct to: SYTO 9.

Lines 189 and 239: You wrote: "at the height of 2 μm" I  think you meant the Z between the captured layers was set to 2 μm. There are some repetition in the second place mentioning this method. Maybe to combine the two techniques into one paragraph.

Line 202: Better to say "washed" rather than "cleaned".

Line 204: The volume of the dilutions seeded on the plates should be stated.

Section 2.8: The source of CV should be stated.

Line 213: Delete "the" before OD.

The formulas used for calculations should be stated.

The composition of buffered peptone water should be stated.

Line 234: Could it be AlexaFluor 647-conjugated dextran 10,000? Please correct.

Line 293 and the whole paragraph: Instead of "accuracy", I would suggest writing "identity".

Figure legend 2: All abbreviations should be spelled out. The concentration labels in subfigure G and H should be made larger and clearer. All concentrations should be used for both strains. And statistics should be added to G and H.

The first sentence in Section 3.2 should be rephrased. It is clumsy.

Line 316: correct to: "concentrations".

When you tested the minimum biofilm inhibitory concentration – from the beginning you have less biofilm of the ASvicK strain. Did you calculate against the untreated ASvicK strain?

The sentence in line 323 should be rephrased. It is better to say that you tested the susceptibility of the two mentioned strains to DMAHDM.

Line 330: The text should be scientific and accurate. So, you should state the percentage of changes.

Line 332: The verb show/depict should be in present tense.

Figure 3: Please explain why the CFU data does not go along with the confocal data.

How can it be both d and e in Figure 3C? State how many times this has been studied. According to confocal – there is a strong reduction in biofilms in both wt and ASvicK. This needs to be commented on.

If you look on the percentage reduction of metabolic activity of wt in the absence/presence of DMAHDM and compare it to the percentage reduction of ASvicK in the absence/presence of DMAHDM, then there is a higher reduction for WT. (Figures 3-5). This suggest that some of the effects of DMAHDM might be by repressing vicK. This has to be tested.

Figure 4: The CV goes along with CFU.

Spelling mistake in line 379 – written WG…

Figure 6: "de" in subfigure A. Only one letter can be used.

Does DMAHDM affect vicK expression in wt?

If the MIC of DMAHDM is 14 μg/ml, you use 6-12.5 times higher concentrations for the biofilm studies. Please comment on it.

Also: In every figure legend dealing with biofilms, you need to state if this is biofilm formation or treatment of preformed biofilms.

Figure 3: Also, the relative SYTO 9, PI intensities should be presented in the graph.

Figure 5: State the units of lactic acid production.

The data in Section 3.6 is relative to control Enamel – so you have to speak about percentage prevention of reduction instead of "fold" increase in hardness. DMAHDM prevents the erosion of the enamel – it does not increase the hardness. So, the text has to be rephrased.

Line 455: Spelling mistake. Correct to "strains".

Line 467: Is there only one or more Ftfs in S. mutans? Make it single/plural accordingly.

Line 480: As discussed above. You can't say "ASvicK-treated biofilms". You also need to provide data showing how much reduction in vicK RNA was achieved with the anti-sense RNA. Maybe DMAHDM further reduces vicK expression. Only knockout can provide an answer for this.

Line 488: For the moment you have only showed that downregulation of vicK in the bacteria using a stable strain was more susceptible to DMAHDM. Therefore, you need to tune down the "method". Only when you have a treatment that enables the introduction of ASvicK into the bacteria of the oral cavity, you can call it a method.

What do you mean by "sound" enamel?

The bottom line: You need to tune down overinterpretations.

Comments on the Quality of English Language

English editing required.

Author Response

(The authors gave the same response as above.)

Round 3

Reviewer 2 Report

Comments and Suggestions for Authors

Spelling mistake in line 64: an e has jumped – correct to “reverse”.

A small question: Would the real-time primers for ASvicK also detect vicK, as it is exactly the same sequence when transcribed into cDNA?

Author Response

(The authors gave the same response as above.)
